# Control of Peri-Implant Mucous Inflammation by Using Chlorhexidine or Ultraviolet C Radiation for Cleaning Healing Abutments. Double-Blind Randomized Clinical Trial

**DOI:** 10.3390/ma13051124

**Published:** 2020-03-03

**Authors:** Arturo Sanchez-Perez, Ana I. Nicolas-Silvente, Carmen Sanchez-Matas, Elena Cascales-Pina, Vanesa Macia-Manresa, Georgios E. Romanos

**Affiliations:** 1D.D.S., Department of Periodontology, Medicine and Dentistry Faculty, Murcia University, 30008 Murcia, Spain; arturosa@um.es (A.S.-P.); elena.cascales1994@gmail.com (E.C.-P.); vanesa.macia@um.es (V.M.-M.); 2D.D.S., Department of Restorative Dentistry, Medicine and Dentistry Faculty, Murcia University, 30008 Murcia, Spain; 3D.D.S., Virgen del Rocio Hospital, 41013 Sevilla, Spain; nem.csm@gmail.com; 4D.D.S., Department of Periodontology, School of Dental Medicine, Stony Brook University, Stony Brook, NY 11794-8712, USA; georgios.romanos@stonybrookmedicine.edu

**Keywords:** abutment surface, healing abutment, mucositis, peri-implantitis, sterilization

## Abstract

Two-phase implants must be exposed to the external environment after the period of osteointegration has elapsed. For this purpose, a healing abutment is placed passing through the mucosa while forming the emergence profile. The continuous connection and disconnection can lead to an alteration in the tissue maturation, both because of the contact of bacterial plaque and because of the mechanical trauma that involves its manipulation, manifesting with different degrees of erythema or bleeding. To assess whether this epithelium disruption can be counteracted, a blinded study design was developed on 150 unitary implant patients divided into three groups (n = 50), applying chlorhexidine (group 1), ultraviolet C (UV-C) at a wavelength of 254 nm (group 2)and no treatment as a control group (group 3), during each of the disconnections and connections during the prosthodontic treatment (1 time per week for four weeks). All groups showed a better epithelium aspect at the end of the evaluation. Although there were no statistically significant differences in the degree of inflammation, the UV-C treated group had the lowest plaque accumulation, and the highest was for the chlorhexidine-treated group.

## 1. Introduction

Dental implants pass through the oral mucosa, establishing a connection between the inner and outer environments [1,2]. Most implants performed today belong to the “two-stage” type. This type of implant involves the appearance of an interface (between the implant and the abutment) called the GAP. When the GAP is positioned at or below the bone level, an increase in initial peri-implant bone loss occurs [3,4,5,6]. This is characteristic for external and internal butt-joint connections but not for Morse tapered (conical) connections, where implants can be placed subcrestally and have long-term crestal bone stability [7,8].

In butt-joint connections, histologically, the process usually manifests itself as an inflammatory state of the implant-abutment complex [9,10], and can trigger a possible loss of crestal bone [11]. One of the possible causes that favor the inflammatory state may be chemical or biological contamination [12,13].

The origin of biological contamination is the oral cavity itself [14], and such contamination may be favored by the initial fear of some patients to brush the newly screwed abutments. In an attempt to prevent this colonization, various strategies have been proposed, ranging from the use of single-phase implants [15,16], sealants, and varnishes [17,18,19], to the careful adjustment of the components through the concept of "GAP 0" [20,21], or modification of the height of the prosthetic abutment to keep it away from the bone crest [22].

Both physical and chemical methods have also been used. Amongst them, the use of irrigations with various solutions has been proposed (water, saline solution or hydrogen peroxide) [23], antiseptics such as chlorhexidine [24], ultrasounds [25], autoclave [26,27], ultraviolet C (UV-C) radiation [28], ozone [23] or cold plasma [2,28].Of all of these, only UV-C, ozone and plasma have shown a good degree of efficacy [23,28]. It should also be noted that most of these studies have been conducted in vitro.

The difficulty for some of these methods to achieve an effective result lies in the strong bond that forms between amino acids and proteins with the titanium surface [29,30,31]. Bacterial contamination of the abutment, therefore, is a phenomenon that is difficult to avoid [32]. Therefore, one-abutment concepts have successfully demonstrated the stability of the bone crest over a long period of time [7].

On the other hand, decontamination and disinfection aim to restore the original composition of the abutments without altering their surface properties or topography, particularly the latter, restricting the use of more expeditious methods [33].

The purpose of this randomized, double-blind clinical trial was to determine whether, in patients receiving two-phase implants, there is an improvement in the degree of mucous inflammation and plaque accumulation over a four-week period, between the connection of the abutment, treated with chlorhexidine, ultraviolet C radiation or with no treatment as a control group.

## 2. Materials and Methods 

### 2.1. Sample Description

#### 2.1.1. Recruitment

Patients who have begun implant treatment (Mozo Grau, Ticare, Valladolid, Spain) and, after two months of tissue healing, are beginning the prosthetic phase of treatment. Data were collected from 18 January 2016 until 24 June 2019. All patients signed a double informed consent, one for implant placement and one for their participation in the study. All patients received detailed information on the purpose of the study and were also informed of the possibility of leaving the study at any time without explanation. The study follows the Helsinki recommendations [34]. The Ethics Committee of the University of Murcia approved the study protocol with ID: 1366/2016. This study was recorded in Clinical Trials (NCT03142828).

#### 2.1.2. Sample Size

A total of 150 patients were recruited into three groups of 50 and were randomly assigned to the test groups or control, which did not differ in treatment indications. 

The size was calculated for the following parameters: confidence level 95%, accuracy 5, approximate value of the value to be measured 3, final sample size of 75 patients, expected loss rate 25%.

The final size obtained by the sample calculation was 97 patients. To fit homogeneous groups, we assumed a final number of 150 patients in 3 groups (n = 50).

#### 2.1.3. Inclusion Criteria

Good systemic health status (ASA I or II), no current pain, no use of painkillers in previous weeks, older than 18 years, an oral hygiene index of < 2 (Löe and Silness), a minimum of 2 mm of attached mucosa, a minimum of 8 mm of vertical bone, a minimum of 7 mm of vestibule-lingual bone, scheduled to receive a unitary implant, and willing to participate in this controlled study.

#### 2.1.4. Exclusion Criteria

Pregnant or women in the lactation period, the use of any medication that might affect the perception of pain, a history of alcohol or drug abuse, a requirement for guided bone regeneration or sinus augmentation procedures and failure to comply with the study protocol.

### 2.2. Study Protocol

Once the implants had been placed and the patients’ consent given, they were treated to connect the healing abutments as they arrived from the manufacturer (Mozo Grau, Ticare, Valladolid, Spain). Patients followed the protocol shown in Figure 1.

Random distribution was made before the assignment, following the Internet program https://www.random.org

The healing abutments were composed of titanium grade 5 (TiAl6V4). Once the abutments had been connected during the second surgery, they were left to heal for one week. After this week, the patients were randomly distributed into three groups: group 1 (G1), application of chlorhexidine gel; group 2 (G2), control using abutment as received from the manufacturer; group 3 (G3), exposure for 10 min in a UV-C sterilizer at a wavelength of 254 nm and 80 W. 

After the first week of healing, the abutments were unscrewed following the sequence: take impressions after a week, testing of the metal structure of the prosthesis, testing of the ceramic prosthesis, and final delivery. The test groups (G1 and G3) received the assigned treatment upon each disconnection, while those of the control group (G2) were kept in a sterile container.

At all times, the data related to the degree of clinical inflammation were recorded. Plaque index was recorded at the end of the evaluation period. A total of four inflammation assessments were performed over four consecutive weeks and a single plaque evaluation at the end of the study period.

Each of the evaluations was interpreted by a blinded evaluator, according to the following evaluation protocol (Figure 2):

The degree of plaque accumulation on the abutment was recorded following the modified Mombelli plaque index [35].

All patients followed CONSORT guidelines to ensure the quality of randomized clinical trials [36].

### 2.3. Randomization

The distribution of patients was carried out using the Internet App (https://www.random.org/), and the result was kept in a closed envelope with the corresponding number from 1 to 150. As patients attended the appointment for their second surgery, the envelope corresponding to their numerical sequence was opened, and the assigned treatment was applied to the abutment. 

### 2.4. Blinding

Patients were informed of the purpose of the study but were unaware of the type of treatment they would receive, and the abutment was stored in a different room when removed for processing, so the group was blinded for the patient. The samples were also blinded to the statistician.

### 2.5. Statistical Analysis

The SPSS version 23.0 statistical package (SPSS Inc., Chicago, IL, USA) was used for statistical comparisons, and an independent statistician reviewed the results. Statistical significance was considered at p < 0.05. 

Data were presented as descriptive statistics (mean and I.C.) using Tukey’s exploratory analysis. Adjustment to normality was determined using the Kolmogorov–Smirnov test, and between-group comparisons were performed using the Mann–Whitney U test and ANOVA test. The results were reviewed by an independent statistician (http://estadisticamurcia.com/web/#2).

## 3. Results

The results are presented as mean and confidence interval (C.I.) for each variable.

A total of 150 patients were included in the study: 62 men and 88 women with an average age of 60.7 years (C.I. 58.9–62.4). The female average age was 60.4 years (C.I. 58.3–62.6), while the male average age was 61.0 years (C.I. 58.0 / 64.0). There were no statistically significant differences between male and female age (t-test).

Twelve implants were placed in anterior position (incisors or canines), 58 in premolars, and 80 in molars. 

Seventeen patients were excluded from the study, with four implant failures, five dropouts, and eight“non-compliance” with appointments. 

Inflammation of the overall sample was 1.3 (C.I. 1.2–1.4) at the time of taking impressions (1 week after connection); 0.9 (C.I. 0.7–1.0) during metal try-in, 0.7 (C.I. 0.6–0.8) during ceramic delivery and 0.5 (C.I. 0.4–0.6) at the time of crown cementation (4 weeks after connection of the abutment), (Figure 2).

The mean amount of plaque after follow-up was 0.7 (C.I. 0.6–0.9).

None of the quantitative variables analyzed met the normality criteria according to the Kolmogorov–Smirnov test.

Results for each group were as follows:Group 1 (treated with chlorhexidine gel) had an average age of 59.3 (C.I. 56.1–62.6), consisting of 29 women with an average age of 59.6 (C.I. 55.2–64.0) and 21 men with an average age of 59.0 (C.I. 53.7–64.3). Inflammation was 1.4 (C.I.1.2–1.6) at the time of impression, 0.8 (C.I. 0.6–1.0) at metal try-in, 0.6 (C.I. 0.4–0.8) in the ceramic delivery and 0.5 (C.I. 0.3–0.7) at the time of prosthesis cementation. The plaque index at the end of monitoring was 0.9 (C.I. 0.7–1.2). Two patients withdrew from the study voluntarily.Group 2 (control without modification) had an average age of 62.2 (C.I. 59.2–65.2), consisting of 27 women with an average age of 62.3 (C.I. 59.4–65.3) and 23 men with an average age of 62.1 (C.I. 56.2–68.0). Inflammation was 1.4 (C.I. 1.1–1.6) at the time of impression, 1.0 (C.I. 0.7–1.2) when metal try-in, 0.8 (C.I. 0.6–1.0) in the ceramic delivery follow up and 0.7 (C.I. 0.5–0.9) at the visit of the prosthesis cementation. The plaque index at the end of monitoring was 0.9 (C.I. 0.6–1.1). There were six dropouts of the study: one missed appointment, two failures, and three withdrawals.Group 3 (treated with UV-C at 254 nm and 80 W) had an average of 60.4 years (C.I. 57.5–63.3), consisting of 32 women with an average age of 59.6 (C.I. 55.8–63.4) and 18 men with an average age of 61.9 (C.I. 57.0–66.8). Inflammation was 1.2 (C.I. 1.0–1.5) at the time of impression, 0.7 (C.I. 0.5–1.0) at the metal try-in, 0.7 (C.I. 0.4–0.9) at the ceramic delivery visit and 0.4 (C.I. 0.3–0.7) at the visit for cementation of the prosthesis. The plaque index at the end of monitoring was 0.5 (C.I. 0.3–0.6). There were nine drop-outs from the study: seven for non-compliance with appointments, and two failures.

No statistically significant differences in the degree of inflammation among the three groups (ANOVA) were found.

The plaque level was statistically significant in favor of group 3 treated with UV-C, with no statistical difference between groups 1 and 2. Clinically, group 1 (chlorhexidine) had the greatest accumulation of plaque (Figure 3).

## 4. Discussion

When connecting an abutment with the implant, there is very often an inflammatory phase, but this is usually limited in severity. Some authors claim that the source of this inflammation is the presence of bacteria inside of the GAP [14,37,38,39]. Such GAP colonization appears to come from the patient’s own oral cavity [14,39,40,41,42], and structured flora has been found in the peri-implant furrow two weeks after connection [43,44]. To reduce contamination of the healing abutment, different compositions have been investigated, such as the use of anatase crystallographic form, with favorable results [45]. In our study, all pillars were grade V (Ti-6Al-4V). So we could not observe this difference. On the other hand, new antiseptics, such as Bactercline® (NMTECH ITALIA SRL), have been developed for decreasing the bacterial charge on the surface of these screws, with hopeful results [46]. In our study, we only use chlorhexidine with no appreciable effect.

In the case of bacterial colonization of the GAP, migration to the inside of the implant seems to depend on the time elapsed since the connection of the abutment. Early initial contamination has been reported after 5 h [47], and 100% of the inside of the implants seems to be contaminated after five years of functioning [13]. This bacterial reservoir can cause bone loss and depends on the so-called “extended arm” [48], "effective radius of action" [49] or "inflammatory front" [50].

Other authors have mentioned that trauma from the continuous connection and disconnection of the abutment could also develop this inflammatory state [51,52,53,54,55]. Finally, for other authors, the mobility of the abutment is a major factor, producing a micro-pumping effect [6,56].

Whatever the case, this inflammatory state can cause bone reabsorption, which will depend on the position of the GAP in the bone [5,57,58].To prevent this adverse effect, the continuity solution between the internal and external environment must be sealed by epithelial and connective adhesion [59,60]. To improve this adhesion, various modifications have been made to the materials of the healing abutments [61,62] and the surface’s topography [63]. However, other authors find no clear improvement from taking such actions [64,65]. To avoid these possible confounding variables, all implants in this study had an internal conical connection, due to their increased sealing characteristics [66]. Likewise, all the healing abutments were used as received from the manufacturer, with a length of 2 mm, and screwed at a standardized torque of 20 N/cm. The only modification they received was that associated with their group treatment. The abutments were screwed in and unscrewed on five time intervals (first connection, impression appointment, metal try-in, ceramic and prosthesis delivery), although they were only evaluated on four occasions to determine inflammatory changes (impression taking, metal test, ceramic test, and delivery).

This implies that the continuous connection and disconnection of the abutments would involve the re-establishment of a new biological width to a more apical level of the implant [51]. Both causes (the connection of the abutment and its colonization) favor the presence of a reservoir between the implant and the abutment [9,62,63,67,68].

To the best of our knowledge, there have been no studies comparing the use of chlorhexidine and UV-C in comparison to untreated controls. In this study, we wanted to determine whether the use of chlorhexidine digluconate gel, the use of UV-C at 254 nm for 15 min or an untreated abutment (as received) lead to differences in the degree of gingival inflammation and plaque accumulation.

As for the degree of clinical inflammation, in our study, there were no significant differences in any of the three groups studied. This induces us to think that the most important factor during this period of what might be termed "prosthetic preparation" is the mechanical trauma involved in the repeated connection and disconnection of the abutments.

During the four weeks of follow-up, all the groups showed gradual improvement due to the maturation of the tissues and reconditioning of the supracrestal insertion in a natural healing process.

These results are consistent with those in the literature that point to the abutment trauma and its continuous changes as a fundamental cause of tissue inflammation [51,69].

As for abutments’ plaque accumulation, many clinicians use chlorhexidine to disinfect prosthetic components and prevent bacterial colonization. However, this procedure remains controversial, and our study indicates clinical differences between the three established groups. The chlorhexidine group showed the worst results, although no statistically significant differences were found with the control group. However, this finding is consistent with the adverse effects described for chlorhexidine, which depend on its substantivity. Indeed, both staining and calcium phosphate crystal deposits increase with the use of chlorhexidine [70]. By contrast, some authors found an improvement in bacterial load with the use of chlorhexidine [71,72]. Our results showed that this decrease in the bacterial load did not result in any clinical improvement during the four weeks of the study. These results agree with those obtained by other authors, who also observed no improvement with the use of chlorhexidine [73,74]. In addition, chlorhexidine may alter the surface properties of titanium, the clinical impact of which has yet to be determined [75].

In the present study, the best result was obtained by the UV-C treated group, which showed statistically significant differences from the other groups regarding plaque accumulation.

The effect of UV-C is complex and is due, on the one hand, to the rejuvenating effect on titanium, eliminating the carbonic compounds deposited on it [76] and, on the other hand, to the alteration caused on the surface load of titanium and its ability to become moist [77]. Finally, UV-C radiation itself is capable of producing ozone, which could also be involved in the disinfection of the abutments [78].

From the clinical standpoint, the stabilized blood clot at the second stage surgery seems to be a physiologic barrier contributing as a fundamental biologic seal to control bacterial penetration to the peri-implant tissues, The clinical relevance of our study is that for the time taken for the tissue to mature, as mentioned above, all groups showed a clear improvement in the degree of inflammation over time, an improvement that was clearly evident two weeks after connection. Therefore, we recommend that, when taking the final impressions, a minimum period of two weeks between the healing abutment connection and the final impression. This aspect is of great significance when the prostheses are located in an aesthetic zone.

## 5. Conclusions

Within the limitations of this study, we can conclude that:No treatment showed a statistically significant decrease in the degree of clinically recorded inflammation.The worst result in terms of plaque accumulation was observed in the chlorhexidine treated group.The best result regarding plaque accumulation was presented by the abutments that were UV-C treated at 254 nm for 15 min, with statistically significant differences.Peri-implant tissue maturation improves with time.Since the most important changes take place during the first two weeks, we recommend taking impressions at this time, although for the best aesthetic results, we recommend waiting four weeks for soft tissue maturation.

## Figures and Tables

**Figure 1 materials-13-01124-f001:**
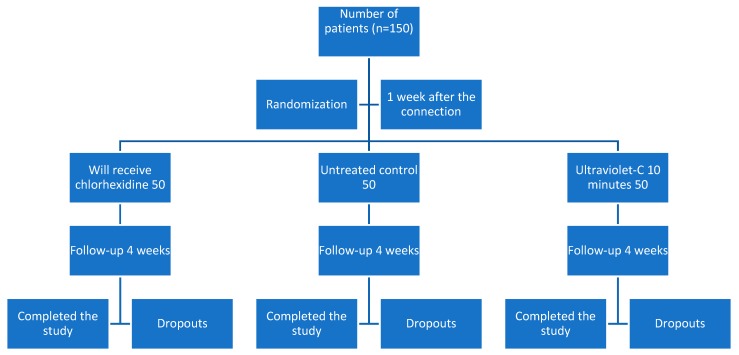
Protocol for distributing the studied sample and assigning them to the test and control group.

**Figure 2 materials-13-01124-f002:**
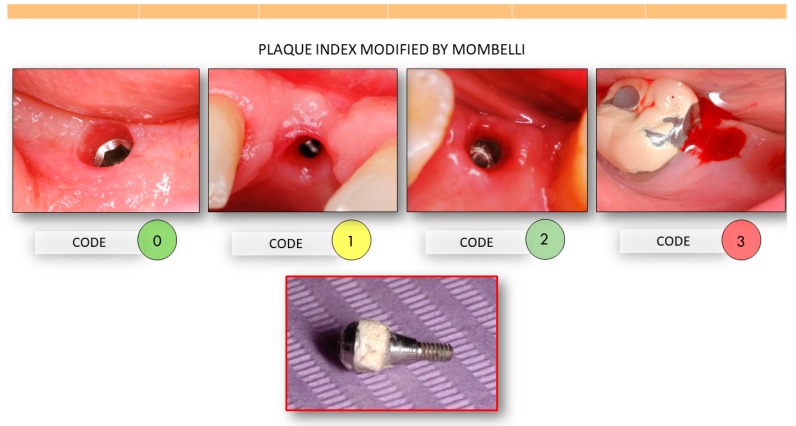
Degree of inflammation of the peri-implant mucosa evaluated immediately after the disengagement of the healing abutment, codified with plaque index modified by Mombelli, being Code 0: gingiva without inflammation and pale pink color; Code 1: gingiva erythematous without bleeding upon manipulation; Code 2: gingiva bleeds slightly during unscrewing or screwing of the abutment; Code 3: gingiva bleeds profusely during unscrewing or screwing of the abutment.

**Figure 3 materials-13-01124-f003:**
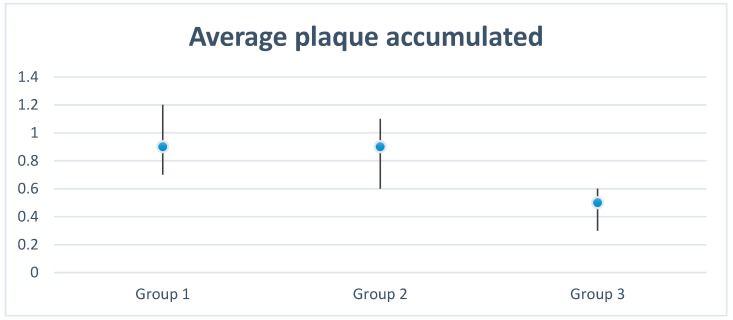
Average plaque accumulated at the end of monitoring on the healing abutments on each experimental group. Group 1: 0.9 (C.I. 0.7–1.2); Group 2: 0.9 (C.I. 0.6–1.1) and Group 3: 0.5 (C.I. 0.3–0.6).

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
