# Peer review of "Control of Peri-Implant Mucous Inflammation by Using Chlorhexidine or Ultraviolet C Radiation for Cleaning Healing Abutments. Double-Blind Randomized Clinical Trial"

_materials, 2020, doi:10.3390/ma13051124_

Round 1
Reviewer 1 Report
Interesting study with a good sample size.
It would have been good to have included more methods to evaluate inflammatory response in the peri-implant mucosa.
I have a couple of comments:
Pg 1 Line 23: Please replace 'emergency' with 'emergence profile'.
Pg 3 Line 98: Please replace 'adhered gum' with 'attached mucosa'.
Methods: Could you please mention the composition of the mealing abutments used in the study.
Author Response
Dear Reviewer,
Thank you for your time and comments that certainly improve the quality and clarity of our work.
Following your recommendations, the following modifications were completed:
Reviewer:
It would have been good to have included more methods to evaluate inflammatory response in the peri-implant mucosa.
Authors:
We totally agree with your comment, but using another method of evaluation would involve the invasion of the peri-implant space and possibly cause bleeding from the surcus.
Undoubtedly, the measurement of peri-implant fluid is much more accurate, but the introduction of the paper strip is a stimulus for bleeding in cases of moderate or severe inflammation.
On the other hand, probing bleeding (BOP) usually gives more false positives in implants than when teeth are explored, due to the lower resistance of the epithelium (see Lang's work), especially when the tissue is inflamed.
Reviewer:
Pg 1 Line 23: Please replace 'emergency' with 'emergence profile'.
Pg 3 Line 98: Please replace 'adhered gum' with 'attached mucosa'.
Authors:
Thank you for that point. We proceed to change.
Reviewer:
Methods: Could you please mention the composition of the mealing abutments used in the study.
Authors:
Healing abutments are composed of titanium grade 5:TiAl6V4. It has been introduced in the text after figure 1.
Reviewer 2 Report
This study was double blinded randomized clinical trials and generally well written.
As each group was independent, connecting line in Fig. 3 seemed to be improper. Bar type graph would be better with detailed explanation about groups and key findings. The numbering style of group seemed to be unfamiliar. Group 1 should be control to avoid any confusion. Line 210, "GROUP 3" was used. However, line 211 "group 1" was used. In the same paper, the style should be consistent. Line 266, "Indeed, both staining and calcium phosphate cystal deposits increase with the use of chlorhexidine." should have proper citation. In addition, typo error is also found. It should be "crystal". Line 272, "In addition, chlorhexidine may alter the surface properties of titanium, the clinical impact of273 which has yet to be determined [73]." Is this sentence enough to make a paragraph? I suggest, this should be followed by upper paragraph. Line 274. This sentence also should be a component of following paragraph, not as independent paragraph.
Author Response
Dear Reviewer,
Thank you for your time and comments that certainly improve the quality and clarity of our work.
Following your recommendations, the following modifications were completed:
Reviewer:
As each group was independent, connecting line in Fig. 3 seemed to be improper. Bar type graph would be better with detailed explanation about groups and key findings.
Authors:
The graphic type has been changed and an explanation has been added.
Reviewer:
The numbering style of group seemed to be unfamiliar. Group 1 should be control to avoid any confusion. Line 210, "GROUP 3" was used. However, line 211 "group 1" was used. In the same paper, the style should be consistent.
Authors:
Thanks for pointing this. Changes in style have been done.
Reviewer:
Line 266, "Indeed, both staining and calcium phosphate cystal deposits increase with the use of chlorhexidine." should have proper citation. In addition, typo error is also found. It should be "crystal".
Authors:
Effectively, there was a typo error that is already corrected. Citation has been also added.
Reviewer:
Line 272, "In addition, chlorhexidine may alter the surface properties of titanium, the clinical impact of
273 which has yet to be determined [73]." Is this sentence enough to make a paragraph? I suggest, this should be followed by upper paragraph. Line 274. This sentence also should be a component of following paragraph, not as independent paragraph.
Authors:
Paragraph changed.
Reviewer 3 Report
Dear Editor,
thank you for giving me the opportunity to review this manuscript.
The topic is interesting and the research well done, moreover there are some minor points which must be revised to make this paper suitable for publication.
1) Results: the paragraphs from line 169 to 176 should moved to the Materials and Methods section.
2) The section "conclusion" is written in a "highlights" style, it should be better to re-write it in the same style used in the other sections.
Best Regards
Author Response
Dear Reviewer,
Thank you for your time and comments that certainly improve the quality and clarity of our work.
Following your recommendations, the following modifications were completed:
Reviewer:
- Results: the paragraphs from line 169 to 176 should moved to the Materials and Methods section.
Authors:
This paragraph has been developed in results because it includes a descriptive statistic of the sample distribution.
Reviewer:
2) The section "conclusion" is written in a "highlights" style, it should be better to re-write it in the same style used in the other sections.
Authors:
Thank you for pointing that. It has been already changed.
Round 2
Reviewer 2 Report
Authors addressed all issues raised by reviewer.
Author Response
Dear reviewer,
We really appreciate your collaboration for the improvement of our manuscript through your review.
Reviewer 3 Report
Dear Editor,
Authors made all the requested changes and now the paper may be considered for publication.
Best regards.
Author Response

(The authors gave the same response as above.)
